# Comparison of Laboratory Data between Children with Kawasaki Disease and COVID-19

**DOI:** 10.3390/children9050638

**Published:** 2022-04-28

**Authors:** Xiao-Ping Liu, Ying-Hsien Huang, Yuh-Chyn Tsai, Shih-Feng Liu, Ho-Chang Kuo

**Affiliations:** 1The Department of Emergency and Pediatrics, Shenzhen Baoan Women’s and Children’s Hospital, Jinan University, Shenzhen 518102, China; yaoguqian@aliyun.com; 2Kawasaki Disease Center and Department of Pediatrics, Kaohsiung Chang Gung Memorial Hospital, Kaohsiung 83301, Taiwan; yhhuang123@yahoo.com.tw; 3College of Medicine, Chang Gung University, Taoyuan 33302, Taiwan; 4Department of Respiratory Therapy, Kaohsiung Chang Gung Memorial Hospital, College of Medicine, Chang Gung University, Kaohsiung 83301, Taiwan; jane2793@cgmh.org.tw; 5Department of Internal Medicine, Division of Pulmonary & Critical Care Medicine, Kaohsiung Chang Gung Memorial Hospital, College of Medicine, Chang Gung University, Kaohsiung 83301, Taiwan

**Keywords:** childhood, COVID-19, Kawasaki disease, Kawasaki-like disease, multisystem inflammatory syndrome in children (MIS-C)

## Abstract

Background: Coronavirus disease 2019 (COVID-19) has been an emerging, rapidly evolving situation in China since late 2019 and has even become a worldwide pandemic. The first case of severe childhood novel coronavirus pneumonia in China was reported in March 2020 in Wuhan. The severity differs between adults and children, with lower death rates and decreased severity for individuals under the age of 20 years. Increased cases of Kawasaki disease (KD) have been reported from New York City and some areas of Italy and the U.K., with almost a 6–10 times increase when compared to previous years. We conducted this study to compare characteristics and laboratory data between KD and COVID-19 in children. Methods: We obtained a total of 24 children with COVID-19 from a literature review and 268 KD cases from our hospital via retrospective chart review. Results: We found that patients with KD have higher levels of white blood cells (WBCs), platelets, neutrophil percentage, C-reactive protein (CRP), procalcitonin, and aspartate aminotransferase (AST) and a higher body temperature, while patients with COVID-19 have a higher age, hemoglobin levels, and lymphocyte percentage. After performing multiple logistic regression analysis, we found that age, WBCs, platelets, procalcitonin, and AST are identical markers for distinguishing COVID-19 from KD in children. Conclusion: In this COVID-19 pandemic period, clinicians should pay attention to children with COVID-19 infection when high WBC, platelet, procalcitonin, and AST values are present in order to provide early diagnosis for KD or multisystem inflammatory syndrome in children (MIS-C).

## 1. Background

The novel coronavirus has been reported as severe acute respiratory syndrome coronavirus 2 (SARS-CoV-2), and the disease it causes has been designated as coronavirus disease 2019 (COVID-19) [1]. So far, according to a systemic review [2], children comprise only 1–5% of the total diagnosed cases of COVID-19. Common presenting symptoms of COVID-19 include cough, pharyngeal erythema, headache, and fever [3]. Because the symptoms or clinical characteristics of patients with COVID-19 and patients with Kawasaki disease (KD) have considerable overlap, identification of laboratory markers that may be indicative of COVID-19 would be highly helpful in an acute care setting.

KD is a systemic vasculitis syndrome with five major clinical criteria of symptoms (the 1–2–3–4–5 rapid memory method), including 1 mouth (fissure lips and/or strawberry tongue), 2 eyes (bilateral non-purulent conjunctivitis), 3 fingers check neck lymph node enlargement (>1.5 cm in diameter, usually unilateral), 4 limbs (induration, redness, and/or desquamation), and 5 days of fever with a polymorphic skin rash [4]. Some patients with KD also have respiratory tract or gastrointestinal tract symptoms, such as cough, rhinorrhea, sore throat, headache, vomiting, diarrhea, abdominal pain, and joint pain [5,6]. The clinical symptoms of COVID-19 somewhat overlap with those of KD. The Bacillus Calmette–Guérin (BCG) vaccine scar reaction was reported with regard to KD mostly in routine vaccination countries such as Taiwan, China, and Japan [7]. Some reports have suggested that BCG vaccination may also play a role in protecting individuals against COVID-19 [8,9].

Multisystem inflammatory syndrome in children (MIS-C) is an emerging disease entity occurring in children with COVID-19 infection. Patients with MIS-C often present with KD-like symptoms. A skin rash over the toe area has been reported by dermatologists, indicating some similar symptoms between COVID-19 and KD in vasculitis [10]. Other symptoms such as the loss of smell or taste in COVID-19 may overlap with hearing loss in patients with KD. The American Academy of Otolaryngology-Head and Neck Surgery (AAO-HNS) proposed that “anosmia could be added to the list of screening tools for possible COVID-19 infection” [11]. The aim of this article is to compare differences between childhood patients with KD and those with COVID-19 to provide clinical information for suspicion of KD.

## 2. Materials and Methods

A retrospective analysis of the medical records of patients admitted to the Shenzhen Baoan Women’s and Children’s Hospital in China was conducted from 2017 to 2021. All the children were under the age of 18 years. A KD diagnosis was provided according to the American Heart Association (AHA) criteria, with at least four of the following five clinical manifestations: bilateral bulbous conjunctival congestion, fissure lips or oral cavity changes, hand and foot symptoms, dysmorphic skin manifestations, and cervical lymph node enlargement [12,13]. Children with incomplete or atypical KD (fever for ≥5 days associated with 2 or 3 of the principal clinical features of KD, *N* = 10) were also enrolled for comparison with those with COVID-19 [14]. We enrolled a total of 24 children (8 + 10 + 6) with age less than 18 years, with a COVID-19 diagnosis, and with laboratory data available online from PubMed [15,16,17]. Laboratory data of the white blood cell/differential count (WBC/DC), C-reactive protein (CRP), procalcitonin (PCT), aspartate aminotransferase (AST), age, and gender were all collected for analysis. The aim of this report was to identify laboratory markers to help clinicians distinguish KD from COVID-19 during this pandemic period and review the disease characteristics. All subjects provided their informed consent for inclusion before they participated in the study.

### Statistics

The data in this study were analyzed with median values with quartiles (Q1, Q3) being used for measurement data and n and percentages used for enumeration data. We adopted the non-parametric Mann–Whitney *U* rank sum test for comparison between the two groups. The chi-square test was performed for countermeasurement. The receiver operating characteristics (ROC) curve method was used to differentiate between groups. The best cut-off points for accurate diagnosis were based on the highest sensitivity value plus the specificity identified by the ROC curve, and the cut-off values were as follows: age, 59 months; WBCs, 9.71 × 10^9^/L; platelets, 273 × 10^9^/L; PCT, 0.13 ng/L; and AST, 21.71U/L. Multivariate logistic regression was performed to analyze the influencing factors that showed significance in univariate analysis. Statistical significance was defined as a *p*-value of <0.05. SPSS (IBM SPSS Statistics for Windows, Version 22.0. Armonk, NY, USA) statistical software was used for all analyses.

## 3. Results

We enrolled 268 patients with KD (258 patients with typical KD and 10 with atypical KD) and 24 patients with COVID-19 for analysis. As shown in Table 1, the lymphocyte percentage (*p* = 0.031) and Hb levels (*p* < 0.001) were significantly higher in patients with COVID-19 than in patients with KD. Meanwhile, the WBC count (*p* < 0.001), neutrophil percentage (*p* < 0.001), platelet count (*p* < 0.001), CRP level (*p* < 0.001), procalcitonin level (*p* < 0.001), and AST level (*p* = 0.015) were significantly lower in patients with COVID-19 compared to patients with KD. Meanwhile, the cut-off values of WBC and CRP levels were 9.71 × 10^9^/L and 38.57 mg/L, respectively. There was no significant difference regarding the levels of ALT between patients with KD and with COVID-19 (*p* = 0.453).

After performing multiple logistic regression, we found that age, WBC counts, platelet counts, procalcitonin levels, and AST levels showed a significant difference between patients with KD and with COVID-19 (*p* < 0.05). The comparison of disease characteristics is shown in Table 2. The etiology of KD remains unknown, but COVID-19 is caused by human coronavirus. Diagnostic criteria for KD include five major symptoms and fever for more than 5 days, but the clinical symptoms or signs of COVID-19 are non-specific or even asymptomatic or without fever. Furthermore, KD has a male-predominant characteristic, and the global prevalence of KD is higher in those of Asian descent than in European or American descent [18], while the prevalence of COVID-19 is higher in the Americas than in Europe and Asia. We also enrolled 10 children with atypical KD for comparison with those with COVID-19 and found significantly higher levels of CRP, WBCs, and procalcitonin in KD than in COVID-19 (*p* < 0.05) and lower hemoglobin in KD (*p* < 0.05). There were 24 patients with KD enrolled for analysis after the COVID-19 epidemic, and no patients with KD (0/24) showed positive results on COVID-19 polymerase chain reaction (PCR).

## 4. Discussion

The COVID-19 pandemic had already caused over 493.6 million infections and over 6,158,000 deaths until April 2022. Although the exact prevalence of COVID-19 in children is still unknown, mounting evidence has highlighted that the clinical severity of COVID-19 in children and young adults appears significantly milder compared to older individuals with comorbidities [21,22]. COVID-19 was also reported to induce Kawasaki-like disease, the multisystem inflammatory syndrome in children (MIS-C), a novel syndrome linked to SARS-CoV-2. In this study, we compared the differences between child patients with COVID-19 (not MIS-C) and with KD (not including KD shock syndrome (KDSS)) and demonstrated that higher WBC, platelet, procalcitonin, and AST levels are laboratory markers for distinguishing COVID-19 from KD. We further revealed that children with KD have a lower age and higher characteristics of fever than children with COVID-19. Other laboratory data, including lactate dehydrogenase (LDH), D-dimer, creatine kinase (CK), and serum creatinine, were also increased in children with COVID-19. According to the statement of the American Heart Association (AHA) [12,23], LDH, D-dimer, CK, and serum creatinine are not on the list of supplementary criteria of suspected incomplete KD. However, LDH, CK, serum creatinine, and D-dimer levels were reported to be higher in patients with KD and associated with IVIG resistance or coronary artery lesion formation [24,25,26,27].

The golden period for IVIG treatment in patients with KD is around 5 to 9 days after disease onset; therefore, early awareness of KD is important for both clinicians and parents. However, the challenge for clinicians in pediatric emergency departments is early identification of KD, because KD shares many clinical signs with other childhood febrile illnesses [28]. Furthermore, with “stay at home” orders and trepidation related to COVID-19 infection, many parents now hesitate or fear seeking in-person consultation for their children. Harahsheh et al. raised the concern of a future surge in the prevalence of CAAs caused by the potential for missed or late diagnosis and treatment of KD in children [29].

In 1974, Tomisaku Kawasaki first described 50 cases of KD in the English language [30]. The etiopathogenesis of KD remains unknown even today, and many studies have failed to identify a pathogen responsible for KD, or any identified pathogens could not be repeated between studies [31]. Furthermore, growing evidence has demonstrated that KD may be the result of a combination of infection, genetics, the environment, and immunity [32]. In addition to standard diagnostic criteria, patients with KD may experience a variety of non-specific clinical features, including uveitis, aseptic meningitis, gastrointestinal symptoms, maculopapular rash, impaired liver function, and anemia [7,33,34]. Hepcidin-induced iron deficiency was reported to be related with transient anemia and disease outcomes in patients with KD, but anemia was not found in children with COVID-19. The neutrophil percentage was higher in patients with KD than in those with COVID-19, while the lymphocyte percentage was higher in children with COVID-19 than in those with KD, indicating a viral immune response in COVID-19.

In the most severe cases of KD, patients may develop hemodynamic instability, known as KD shock syndrome (KDSS), and secondary hemophagocytic lymphohistiocytosis fulfils the criteria of macrophage activation syndrome (MAS) [35]. Surprisingly, several lines of study have suggested that children with COVID-19 were significantly unwell across Europe and the United States, with a novel syndrome linked to SARS-CoV-2 (MIS-C) [36,37]. In this rare syndrome of COVID-19, children share common features with other pediatric inflammatory conditions, including KD, staphylococcal/streptococcal toxic shock, macrophage activation syndrome, and sepsis [37].

In 2005, Esper et al. identified a novel human coronavirus tested by RT-PCR, designated New Haven coronavirus (HCoV-NH), in the respiratory secretions of 8 of 11 children with KD compared to 1 of 22 controls [38]. Notably, Jones et al. reported a 6-month-old infant diagnosed with KD and treated with IVIG and aspirin and a positive screening of COVID-19 in Italy [39]. Thereafter, Verdoni et al. reported a case series of a 30-fold increased incidence of Kawasaki-like disease at the Italian epicenter of the COVID-19 pandemic [40]. Patients who showed evidence of an immune response to the COVID-19 virus were older, had a higher rate of cardiac involvement, and had macrophage activation syndrome features [40]. Consistently, in Whittaker et al.’s study, pediatric patients with MIS-C were generally older than those with KD or KDSS and had higher WBC and CRP levels, as well as more profound lymphopenia and anemia [41]. The COVID-19 pandemic has also been associated with a high incidence of a severe form of KD [42]. The authors also identified that the Kawasaki-like disease described here remains a rare condition, probably affecting no more than 1 in 1000 children exposed to COVID-19 [40]. The limited number of children with COVID-19 (*N* = 24) included in this study is a limitation and weakness of this study. However, the association between KD and COVID-19 still warrants further investigation. In the era of the COVID-19 pandemic, multisystem inflammatory presentations of children with COVID-19 and patients with typical KD will be more challenging for clinicians and pediatricians in the future.

## 5. Conclusions

In conclusion, a younger age, male gender, and higher levels of WBCs, platelets, procalcitonin, and AST are identical markers for distinguishing pediatric patients with KD from those with COVID-19 and not MIS-C. In this period of the COVID-19 pandemic, clinicians should pay more attention to children with COVID-19 infection and laboratory data showing high WBC, platelet, procalcitonin, and AST levels to survey the possibility of KD. These laboratory markers were ancillary findings in patients with COVID-19 compared to KD. If a patient with COVID-19 has these markers, which are also suggestive of KD, evaluation of the clinical major symptoms of KD may be considered as well. Likewise, children with KD should also be screened for COVID-19 infection.

## Figures and Tables

**Table 1 children-09-00638-t001:** Comparison of laboratory values between children with COVID-19 and with Kawasaki disease.

	COVID-19 (*N* = 24)	Kawasaki Disease (*N* = 258)	Univariate *p*-Value	Multivariate *p*-Value	Multivariate (B)	Odds Ratio	95% CI	
							Lower	Upper
Age (months)	54 (16, 105)	20 (12, 35.2)	<0.001	0.012	−1.965	0.140	0.030	0.651
Male gender (%)	12 (50%)	152 (58.9%)	0.397	-				
White blood cell count (×10^9^/L)	6.59 (3.87, 9.57)	14.1 (10.1, 17.5)	<0.001	0.025	1.858	6.409	1.267	32.415
Neutrophil %	47.51 (28.59, 58.8)	61.5 (50, 73.7)	<0.001	-				
Lymphocyte %	35.31 (20.5, 50.88)	26.6 (17.1, 38.2)	0.031	-				
Platelet count (×10^9^/L)	207 (156.75, 301.25)	350 (268, 448)	<0.001	0.03	1.728	5.627	1.178	26.886
Hemoglobin (g/L)	122 (113, 134.75)	108 (100, 115)	<0.001	-				
C-reactive protein (mg/L)	10.85 (5.82, 30)	64.8 (34.1, 111.9)	<0.001	-				
Procalcitonin (ng/L)	0.07 (0.048, 0.095)	0.44 (0.16, 1.31)	<0.001	0.001	2.890	17.985	3.219	100.493
AST (U/L)	31.5 (20.35, 40)	36 (29, 52)	0.015	0.009	3.150	23.333	2.166	251.365
ALT (U/L)	18.5 (13.7, 42.25)	21 (13, 49)	0.453	-				

Data were presented as medians with quartiles (Q1, Q3). AST: aspartate aminotransferase; ALT: alanine aminotransferase; multivariate (B): coefficients of variables in multivariate analysis; OR: odds ratio; CI: confidence interval. The male gender was analyzed by Pearson’s chi-square test, and others were analyzed by the Mann–Whitney *U* test.

**Table 2 children-09-00638-t002:** Different characteristics between Kawasaki disease (KD) and COVID-19.

	Kawasaki Disease (KD)	COVID-19
Etiology	Unknown (coronavirus may be one of the triggers of KD)	Human corona virus
Symptoms	5 major symptoms (fissure lips and/or strawberry tongue, bilateral non-purulent conjunctivitis, neck lymphadenopathy, limb induration, and polymorphic skin rash)	Upper respiratory tract symptoms (non-specific or even asymptomatic)
Fever (>38 °C)	100%	60–70%
Treatment	IVIG + aspirin (steroid for high-risk group)	Anti-IL6, hydroxychloroquine, remdesivir, monoclonal antibodies, baricitinib, steroids, etc.
Age	85% < 5 years old	2% < 19 years old
Gender	Male > female, 1.5-fold	Male = female
BCG vaccine	Scar induration	May play a protective role
Abdominal pain	35% [19]	68% in MIS-C [20] 21% in COVID-19
Prevalence	Asia > Americas > Europe	Europe, Americas > Asia

°C: centigrade (body temperature); IVIG: intravenous immunoglobulin; IL6: interleukin 6; BCG: Bacillus Calmette–Guérin; MIS-C: multisystem inflammatory syndrome in children. Five major symptoms (1–2–3–4–5) of Kawasaki disease: 1 mouth (fissure lips and/or strawberry tongue), 2 eyes (bilateral non-purulent conjunctivitis), 3 fingers to check neck lymph node enlargement (neck lymphadenopathy), 4 limbs (induration or desquamation), and 5 days of fever with a polymorphic skin rash.

## Data Availability

The dataset containing results from this article are available from the corresponding author (H.-C.K.) upon request.

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
