# Peer review of "Comparison of Laboratory Data between Children with Kawasaki Disease and COVID-19"

_children, 2022, doi:10.3390/children9050638_

Round 1
Reviewer 1 Report
This study compared the clinical presentation of Kawasaki disease and COVID-19. This study is of some significance because the association between Kawasaki disease and COVID-19 has recently been attracting attention.
However, I believe some improvements can be made.
Major points
- There is no mention of multisystem inflammatory syndrome in children (MIS-C) in the Introduction. If this subject is to be addressed, this point should be mentioned.
- The study period of this study includes patients with Kawasaki disease from before the COVID-19 epidemic. The time period should be aligned for a proper comparison.
- The clinical picture summary does not include abdominal pain or other symptoms seen in MIS-C. If the authors can find out, it would be useful information.
- Were any patients diagnosed with both Kawasaki disease and COVID-19? This would also be important information.
Author Response
Major points
- There is no mention of multisystem inflammatory syndrome in children (MIS-C) in the Introduction. If this subject is to be addressed, this point should be mentioned.
--> We have added MIS-C in the introduction.
- The study period of this study includes patients with Kawasaki disease from before the COVID-19 epidemic. The time period should be aligned for a proper comparison.
-->We have enrolled 24 KD patients after the COVID-19 epidemic for analysis and added in the result section as well as table.
- The clinical picture summary does not include abdominal pain or other symptoms seen in MIS-C. If the authors can find out, it would be useful information.
--> We have added the abdominal pain for comparison.
- Were any patients diagnosed with both Kawasaki disease and COVID-19? This would also be important information.
-->We have added in the result section and table as “There were 24 KD patients enrolled for analysis after the COVID-19 epidemic, and there were no KD patients (0/24) showed positive results of COVID-19 PCR reaction.”

Reviewer 2 Report
Need to delineate further that this is an evaluation of COVID 19 infection and not MIS-C or if this is truly a way to predict which kids have Kawasaki and which have MIS-C. Distinguish between an acute COVID 19 infection and antibody related prior COVID infection with current inflammation (MIS-C)
lines 62-65 discussing BCG vaccine seem to have little relevance.
table 2 treatment for COVID is outdated
line 149-150 needs clarification about incomplete KD
Author Response
Reviewer 2:
Need to delineate further that this is an evaluation of COVID 19 infection and not MIS-C or if this is truly a way to predict which kids have Kawasaki and which have MIS-C. Distinguish between an acute COVID 19 infection and antibody related prior COVID infection with current inflammation (MIS-C)
--> We have revised in conclusion that "provide identical markers for distinguishing KD patients from COVID-19 in children and not MIS-C ".
lines 62-65 discussing BCG vaccine seem to have little relevance.
--> we have deleted the sentence about BCG.
table 2 treatment for COVID is outdated
--> We have revised it.
line 149-150 needs clarification about incomplete KD
--> we have revised it.

Reviewer 3 Report
In their paper, "Comparison laboratory data between children with Kawasaki 2 Disease and COVID-19", Liu et al. compare characteristics and laboratory data between KD and COVID-19 in children. They conclude that clinicians should pay attention to children with COVID-19 infection when high WBC, platelet, procalcitonin, and AST values are present in order to provide precision treatment with IVIG. I have a few concerns with the study design.
- While the evaluation of laboratory markers in KD and Covid-19 infection may be useful as an adjunctive tool, it should never be used to guide therapy. KD is a clinical diagnosis as the authors have rightly mentioned in their paper. The authors have excluded incomplete KD, where laboratory markers are used as ancillary tests. I do not agree with the conclusion and the message of the paper which states that laboratory data showing high WBC, platelet, procalcitonin, and AST levels should guide treatment with IVIG for possible KD. Again, while these lab markers may raise suspicion, they should not be used to guide therapy. In addition, the clinicians would know if the patient has Covid-19 because they would be tested for it. So although it is interesting to review the data retrospectively, I struggle to see the real-world use of this.
- The message of the paper and the conclusion needs to be revised. The authors may mention that these labs markers are ancillary findings in patients with Covid-19 compared to KD. In addition, if a patient with Covid-19 has these markers which are also suggestive of KD, may consider evaluation for KD as well. It may be more useful to compare patients with incomplete KD and Covid-19 rather than excluding them. These patients are more of a diagnostic dilemma, and it may be worthwhile looking into this cohort.
Author Response
Reviewer 3:
In their paper, "Comparison laboratory data between children with Kawasaki 2 Disease and COVID-19", Liu et al. compare characteristics and laboratory data between KD and COVID-19 in children. They conclude that clinicians should pay attention to children with COVID-19 infection when high WBC, platelet, procalcitonin, and AST values are present in order to provide precision treatment with IVIG. I have a few concerns with the study design.
- While the evaluation of laboratory markers in KD and Covid-19 infection may be useful as an adjunctive tool, it should never be used to guide therapy. KD is a clinical diagnosis as the authors have rightly mentioned in their paper. The authors have excluded incomplete KD, where laboratory markers are used as ancillary tests. I do not agree with the conclusion and the message of the paper which states that laboratory data showing high WBC, platelet, procalcitonin, and AST levels should guide treatment with IVIG for possible KD. Again, while these lab markers may raise suspicion, they should not be used to guide therapy. In addition, the clinicians would know if the patient has Covid-19 because they would be tested for it. So although it is interesting to review the data retrospectively, I struggle to see the real-world use of this.
-->We have revised it. "The aim of this article is to compare differences between childhood patients with KD and COVID-19 to provide clinical information for suspect of KD.
and in the conclusion, we revised as " In this pandemic period of COVID-19, clinicians should pay more attention to children with COVID-19 infection while laboratory data showing high WBC, platelet, procalcitonin, and AST levels to survey the possibility of KD."
- The message of the paper and the conclusion needs to be revised. The authors may mention that these labs markers are ancillary findings in patients with Covid-19 compared to KD. In addition, if a patient with Covid-19 has these markers which are also suggestive of KD, may consider evaluation for KD as well. It may be more useful to compare patients with incomplete KD and Covid-19 rather than excluding them. These patients are more of a diagnostic dilemma, and it may be worthwhile looking into this cohort.
--> We have added into the conclusion section.

Round 2
Reviewer 1 Report
Thank you for correcting the paper.
I have no other comments to add.
Author Response
Thanks a lot.
Reviewer 2 Report
improved section on MIS vs acute COVID
Author Response
Thanks a lot.
Reviewer 3 Report
I thank the authors for making the revisions and the conclusion of the paper reads better. I would recommend clarifying what the authors mean by 'consider evaluation for KD as well.' Do they mean echocardiograms should be performed on all these patients?
Finally, I do think that atypical KD needs to be included in this cohort. I would recommend performing a subset analysis with these patients if possible.
Author Response
1. I thank the authors for making the revisions and the conclusion of the paper reads better. I would recommend clarifying what the authors mean by 'consider evaluation for KD as well.' Do they mean echocardiograms should be performed on all these patients?
--> We have revised it as "evaluation the clinical major symptoms of KD as well"
2. Finally, I do think that atypical KD needs to be included in this cohort. I would recommend performing a subset analysis with these patients if possible.
--> We have added 10 atypical KD for analysis and showed in the result section.
Round 3
Reviewer 3 Report
Thank you to the authors for making the changes. Since atypical KD has now been included, the methods stating that atypical KD has been excluded needs to be changed.
Thank you again for providing me with the opportunity of reviewing this manuscript.
Author Response
We have revised the method section.